# Little evidence for an effect of smoking on multiple sclerosis risk: A Mendelian Randomization study

Ruth E. Mitchell[1,2☯]*, Kirsty Bates[2☯], Robyn E. Wootton[1,3,4], Adil Harroud[5,6], J. Brent Richards[7,8,9,10,11], George Davey Smith[1,2], Marcus R. Munafò[1,4,12]

**1** Medical Research Council Integrative Epidemiology Unit, University of Bristol, Bristol, United Kingdom, **2** Population Health Sciences, Bristol Medical School, University of Bristol, United Kingdom, **3** Avon & Wiltshire Mental Health Partnership Trust, Bristol, United Kingdom, **4** School of Psychological Science, University of Bristol, Bristol, United Kingdom, **5** Department of Neurology, University of California, San Francisco, California, United States of America, **6** Weill Institute for Neurosciences, University of California, San Francisco, California, United States of America, **7** Department of Human Genetics, McGill University, Montreal, Quebec, Canada, **8** Centre for Clinical Epidemiology, Department of Epidemiology, Lady Davis Institute for Medical Research, Jewish General Hospital, McGill University, Montreal, Quebec, Canada, **9** Department of Medicine, McGill University Montreal, Quebec, Canada, **10** Department of Human Genetics, McGill University, Montreal, Quebec, Canada, **11** Department of Twin Research and Genetic Epidemiology, King's College London, United Kingdom, **12** NIHR Biomedical Research Centre at the University Hospitals Bristol NHS Foundation Trust and the University of Bristol, Bristol, United Kingdom

☯ These authors contributed equally to this work.
* r.mitchell@bristol.ac.uk

**Data Availability Statement:** Source data for this study comes from summary statistics from genome-wide association studies (GWAS) that are available for download online. Lifetime smoking

## Abstract

The causes of multiple sclerosis (MS) remain unknown. Smoking has been associated with MS in observational studies and is often thought of as an environmental risk factor. We used two-sample Mendelian randomization (MR) to examine whether this association is causal using genetic variants identified in genome-wide association studies (GWASs) as associated with smoking. We assessed both smoking initiation and lifetime smoking behaviour (which captures smoking duration, heaviness, and cessation). There was very limited evidence for a meaningful effect of smoking on MS susceptibility as measured using summary statistics from the International Multiple Sclerosis Genetics Consortium (IMSGC) meta-analysis, including 14,802 cases and 26,703 controls. There was no clear evidence for an effect of smoking on the risk of developing MS (smoking initiation: odds ratio [OR] 1.03, 95% confidence interval [CI] 0.92–1.61; lifetime smoking: OR 1.10, 95% CI 0.87–1.40). These findings suggest that smoking does not have a detrimental consequence on MS susceptibility. Further work is needed to determine the causal effect of smoking on MS progression.

## Background

Smoking is an avoidable environmental cause to many life-threatening diseases such as lung cancer and heart and respiratory disorders [1,2]. There is emerging evidence linking cigarette smoke to conditions negatively affecting the central nervous system (CNS), like multiple sclerosis (MS) [3,4]. MS is a chronic neurological disorder causing autoimmune breakdown of the

GWAS summary data is available at https://doi.org/10.5523/bris.10i96zb8gm0j81yz0q6ztei23d and smoking initiation GWAS summary data is available at https://genome.psych.umn.edu/index.php/GSCAN. Multiple Sclerosis GWAS summary data is available from the International Multiple Sclerosis Genetics Consortium (IMSGC) upon person validation and agreement not to distribute data to third parties at https://imsgc.net/?page_id=31.

**Funding:** REM, REW, GDS and MRM are all members of the MRC Integrative Epidemiology Unit at the University of Bristol funded by the MRC: http://www.mrc.ac.uk (MC_UU_00011/1, MC_UU_00011/7). This study was supported by the National Institute for Health Research (NIHR) Biomedical Research Centre at the University Hospitals Bristol National Health Service (NHS) Foundation Trust and the University of Bristol. The views expressed in this publication are those of the authors and not necessarily those of the NHS, the National Institute for Health Research or the Department of Health and Social Care. AH is funded by the NMSS-ABF Clinician Scientist Development Award from the National Multiple Sclerosis Society (NMSS) and the Multiple Sclerosis Society of Canada (MSSC). AH is funded by the NMSS-ABF Clinician Scientist Development Award from the National Multiple Sclerosis Society (NMSS) and the Multiple Sclerosis Society of Canada (MSSC). JBR is supported by the Canadian Institutes of Health Research (CIHR), the Canadian Foundation for Innovation and the Fonds de Recherche du Québec – Santé (FRQS). JBR is funded by a FRQS Clinical Research Scholarship and received research support from the NMSS and the MSSC. The funders had no role in study design, data collection and analysis, decision to publish, or preparation of the manuscript.

**Competing interests:** The authors have declared that no competing interests exist.

**Abbreviations:** CI, confidence interval; CNS, central nervous system; EDSS, Expanded Disability Status Scale; GWAS, genome-wide association study; IMSGC, International Multiple Sclerosis Genetics Consortium; InSIDE, INstrument Strength Independent of Direct Effect; IVW, inverse-variance weighted; MHC, major histocompatibility complex; MR, Mendelian randomization; MRI, magnetic resonance imaging; MS, multiple sclerosis; NAT1, N-acetyltransferase 1; OR, odds ratio; PRESSO, pleiotropy residual sum and outlier; RAPS, robust adjusted profile score; SD, standard deviation; SNP, single nucleotide polymorphism.

myelin sheath surrounding axons in the CNS [5]. The disease is characterised by periods of disease activity followed by remission and/or progressive neurological decline, resulting in increasing disability [6]. Like most autoimmune conditions, there is no known specific cause; however, we know there is an interaction between genetic and environmental factors in susceptible individuals that go on to develop the disorder [7]. Unfortunately, there is no cure for MS [8], and people diagnosed with MS often live with extreme disability [9]. There are emerging treatments aimed at modifying the disease course [10], but they are not universally effective particularly with regard to the progressive form of the disease. Therefore, it is important to continue targeting prevention by means of establishing causal links.

Evidence from observational epidemiological studies suggests that smoking increases MS risk [11]. It is hypothesised from experimental studies that exposure to chemicals in cigarette smoke alters the immune cell balance in the lung [12,13], which, in turn, can lead to generalised pro-inflammatory effects that trigger autoimmunity [14,15], in genetically susceptible individuals [16,17]. In addition, cigarette chemicals may contribute mechanistically to MS pathobiology. Specifically, nicotine is suggested to increase the permeability of the blood–brain barrier [18], cyanide may contribute to demyelination [19], and nitric oxide could cause degeneration of axons [20]. There is evidence for an association between smoking and worsening symptoms, number of relapses, lesion load on magnetic resonance imaging (MRI), brain atrophy rate [15], and the rapidity of disability progression in MS patients [4,21,22].

However, it is hard to make causal inferences from observational studies, which can be biased by issues of reverse causation and residual confounding. One method which can be used to reduce these sources of bias is Mendelian randomization (MR) [23]. MR can be implemented through instrumental variable analysis that uses genetic variants to proxy the exposure (e.g., smoking) and estimate a causal effect of that exposure on the outcome (e.g., MS). The MR method makes 3 important assumptions: (1) the genetic variants must robustly predict the exposure; (2) the genetic variants must not be associated with any confounders; and (3) the genetic variants must only affect the outcome through the exposure [24]. To satisfy the first assumption, we selected the most recently available genetic instruments from previously conducted genome-wide association studies (GWASs) associated with smoking behaviour (smoking initiation [25] and lifetime smoking [26]) that can be implemented in a two-sample MR context (Fig 1A). The latter 2 assumptions can be violated by horizontal pleiotropy, which occurs when the genetic variants affect the outcome other than through the exposure. We test for this possibility using multiple sensitivity analyses. In order to examine the association between smoking and MS, we chose to investigate smoking behaviour using 2 specific phenotypes relating to the initiation and a lifetime use of tobacco. Smoking initiation indicates whether an individual had ever smoked regularly and the lifetime smoking exposure which captures both smoking initiation (i.e., ever and never smokers) and, among ever smokers, takes into account smoking duration, heaviness, and cessation.

# Results

## Smoking initiation

The inverse-variance weighted (IVW) MR estimate (odds ratio [OR] 1.03, 95% confidence interval [CI] 0.92 to 1.16) revealed no strong evidence for a causal effect of the genetic risk of smoking initiation on the incidence of MS (Fig 2). This was consistent across all MR methods employed, providing further support for the result as each MR method has different assumptions and therefore tests for different violations of those assumptions. Indeed, the weighted median and weighted mode only allow single nucleotide polymorphisms (SNPs) in the largest homogeneous cluster to contribute to the overall estimate and provide estimates with CIs

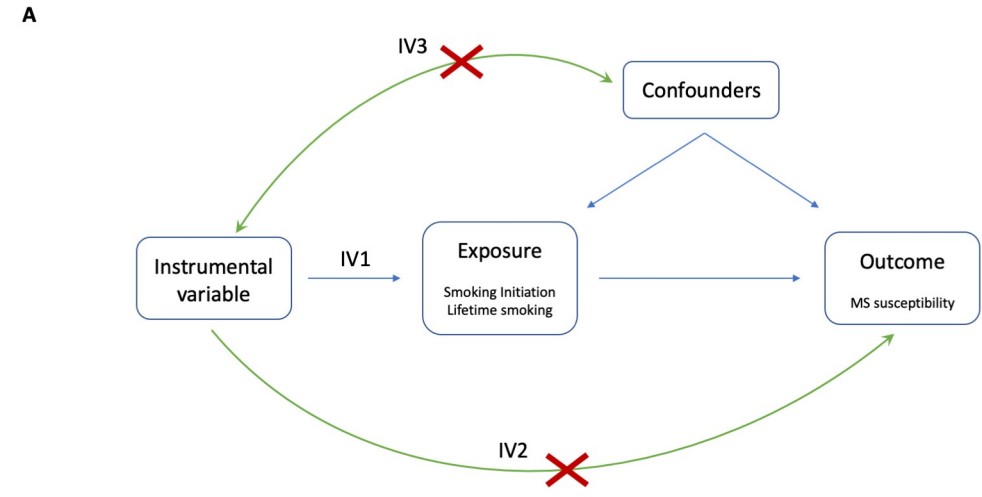

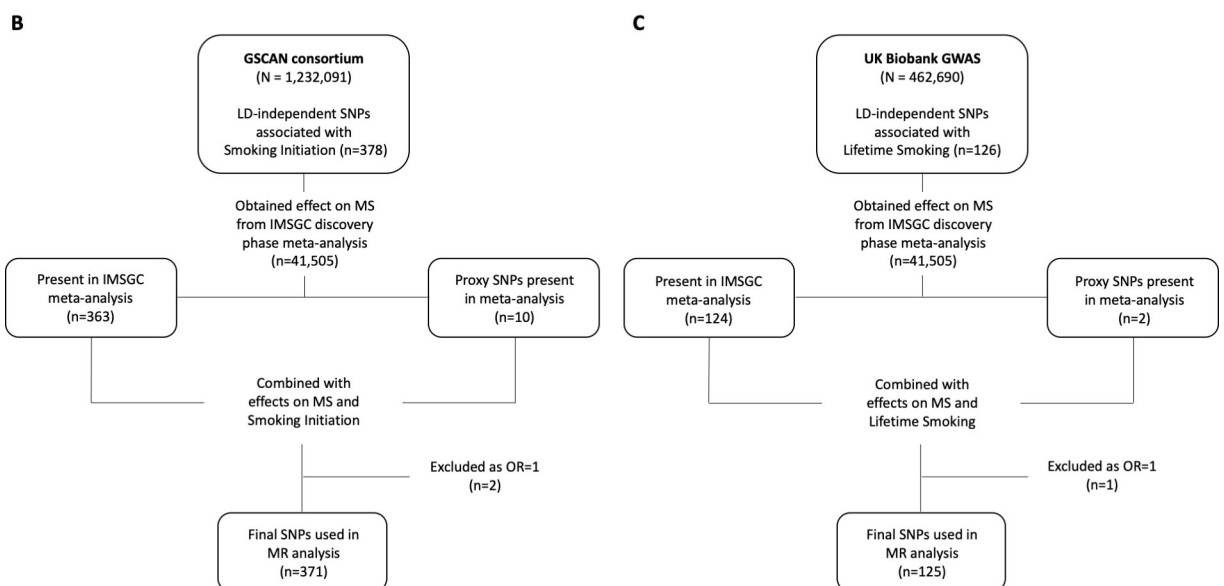

**Fig 1. Schematic of MR analysis.** (A) Directed acyclic graph of the MR framework investigating the causal relationship between smoking and MS. Instrumental variable assumptions: (1) the instruments must be associated with the exposure; (2) the instruments must influence MS only through smoking; and (3) the instruments must not associate with measured or unmeasured confounders in the smoking to MS relationship. (B and C) Flowchart for selection of genetic variants associated with smoking initiation (B) and lifetime smoking (C). GSCAN, GWAS & Sequencing Consortium of Alcohol and Nicotine use; GWAS, genome-wide association study; IMSGC, International Multiple Sclerosis Genetics Consortium; IV, instrumental variable; LD, linkage disequilibrium; MR, Mendelian randomization; MS, multiple sclerosis; OR, odds ratio; SNP, single nucleotide polymorphism.

overlapping the null (Fig 2 and Fig A in S1 Data). The 371 SNPs used as genetic proxies for smoking initiation (Fig 1B and Table A in S1 Data) had an F statistic of 44.90, indicating a strong instrument and that weak instrument bias was unlikely to be influencing the effect estimates. There was evidence of heterogeneity with a large Cochran Q statistic of 559.48, $p = 6.65 \times 10^{-6}$ and the MR-pleiotropy residual sum and outlier (PRESSO) global test value of 562.12, $p < 0.000125$. However, this not indicative of directional horizontal pleiotropy given the consistent MR–Egger estimate (OR 1.13, 95% CI 0.67 to 1.91), small intercept (0.0017,

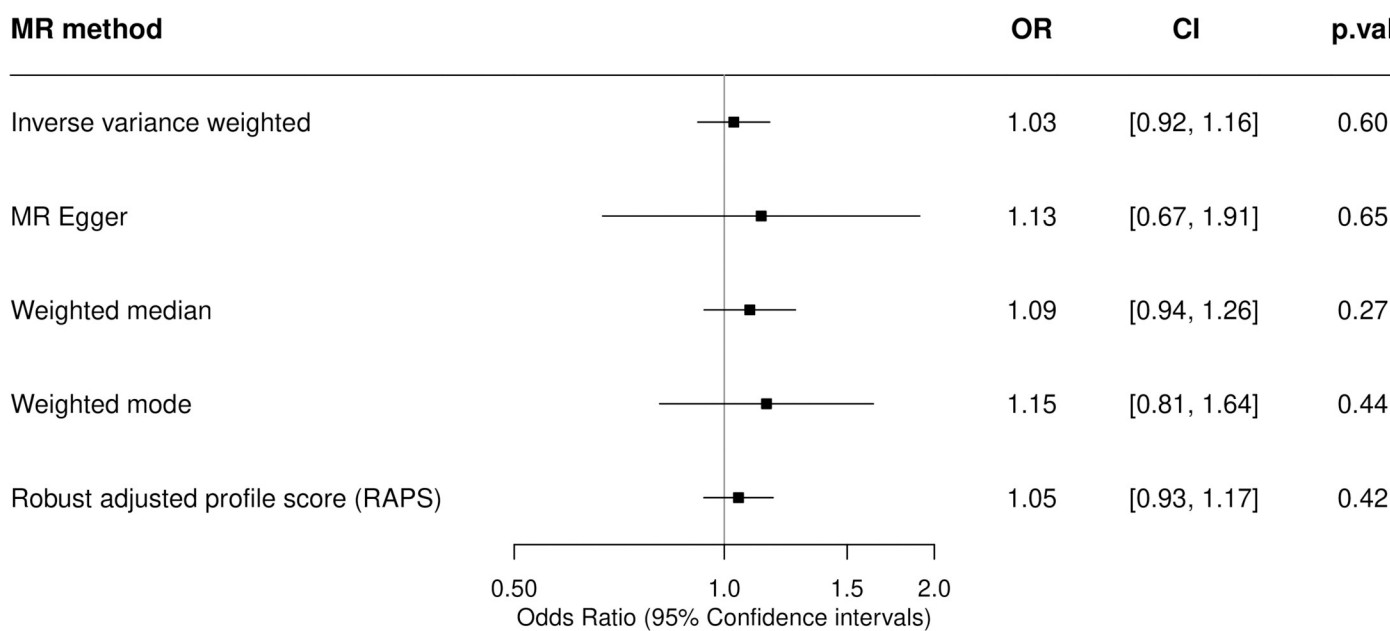

**Fig 2. Two-sample MR estimates of the association between smoking initiation and the incidence of MS.** A two-sample MR analysis was undertaken to obtain causal estimates of genetically predicted smoking initiation on MS susceptibility. MR and sensitivity analyses were performed using the TwoSampleMR R package with a comparison across 5 different methods. ORs are expressed per unit increase in log odds of ever smoking regularly (smoking initiation), with a 1 SD increase in genetically predicted smoking initiation corresponding to a 10% increased risk of smoking. The genetic variants used to proxy smoking initiation are the conditionally independent genome-wide significant SNPs taken from the GSCAN consortium detailed in Table A in S1 Data. The estimates of their association with MS are taken from the 2019 MS Chip IMSGC meta-analysis. CI, confidence interval; GSCAN, GWAS & Sequencing Consortium of Alcohol and Nicotine use; IMSGC, International Multiple Sclerosis Genetics Consortium; MR, Mendelian randomization; MS, multiple sclerosis; OR, odds ratio; p.val, *p*-value; RAPS, robust adjusted profile score; SD, standard deviation; SNP, single nucleotide polymorphism.

*p* = 0.73), and symmetrical funnel plot (Fig B in S1 Data). Similarly, MR-robust adjusted profile score (RAPS) is robust to systematic and idiosyncratic pleiotropy, accounting for weak instruments, pleiotropy, and extreme outliers and gave a similar causal estimate (OR 1.05, 95% CI 0.93 to 1.17). Furthermore, MR-PRESSO removes individual outlier SNPs that contribute to heterogeneity disproportionately in order to correct for horizontal pleiotropy. The MR-PRESSO outlier corrected causal estimate was 1.040 (95% CI 1.040 to 1.041). Therefore, the second instrumental variable assumption (known as the exclusion restriction assumption) of MR has not been violated, and directional pleiotropy is unlikely to be biasing the estimates, even though the outlier removal automatically leads to over precise estimates. Leave-one-out and single-SNP analyses (Fig C and D in S1 Data) were conducted as sensitivity tests sequentially omitting 1 SNP at a time and performing MR using a single SNP, respectively, to assess the sensitivity of the results to individual variants. These indicated that there is not a single SNP driving the association whose effect is being masked in the overall analysis. The exclusion of exposure variants located within the major histocompatibility complex (MHC) did not alter the null association between smoking initiation and the incidence of MS (Table B in S1 Data).

## Lifetime smoking

There was no clear evidence for a causal effect of the genetic risk of lifetime smoking on the incidence of MS (Fig 3). The 125 SNPs used as genetic proxies for lifetime smoking (Fig 1C and Table C in S1 Data) had an F statistic of 44.05, indicating a strong instrument that is unlikely to cause the effect estimates to be affected by weak instrument bias. The IVW MR analysis estimate (OR 1.10, 95% CI 0.87 to 1.40) revealed no strong evidence for a causal effect of the genetic risk of lifetime smoking on the incidence of MS and was consistent across all

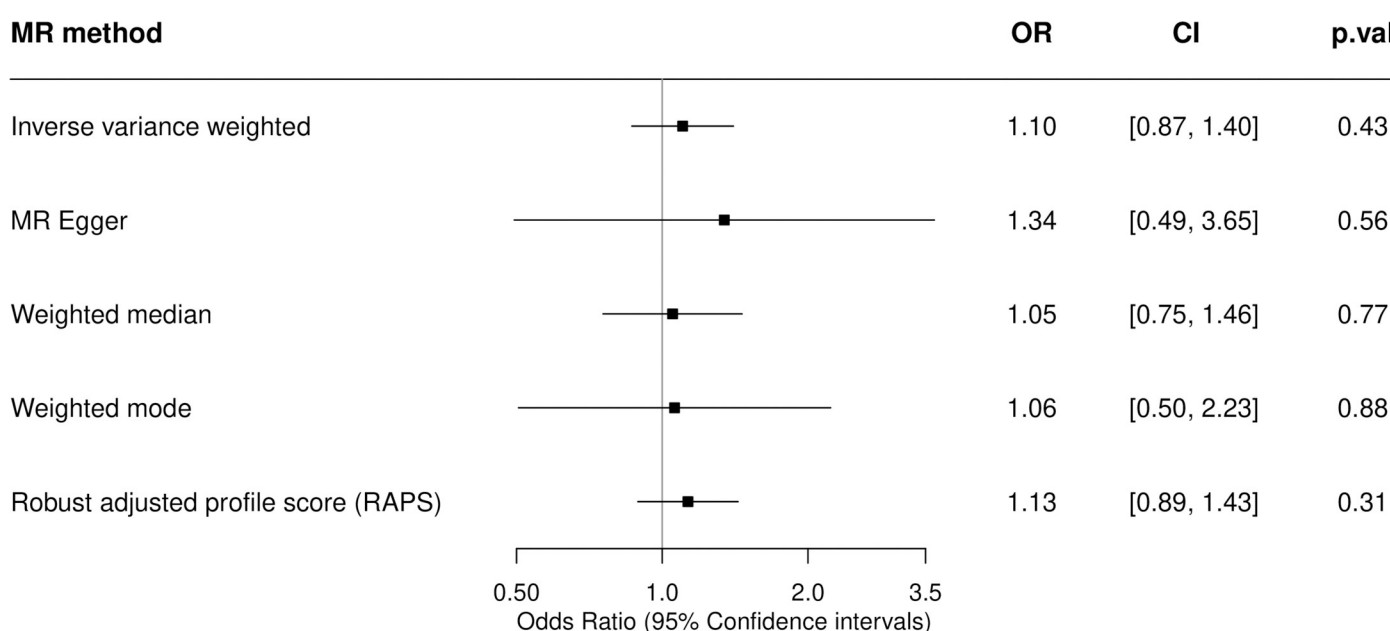

**Fig 3. Two-sample MR estimates of the association between lifetime smoking and the incidence of MS.** A two-sample MR analysis was undertaken to obtain causal estimates of genetically predicted lifetime smoking on MS susceptibility. MR and sensitivity analyses were performed using the TwoSampleMR R package with a comparison across 5 different methods. ORs are expressed per 1 SD increase of the lifetime smoking index. A SD increase in the lifetime smoking score is equivalent to an individual smoking 20 cigarettes a day for 15 years and stopping 17 years ago or an individual smoking 60 cigarettes a day for 13 years and stopping 22 years ago. The genetic variants used to proxy lifetime smoking are the independent genome-wide significant SNPs taken from the GWAS of lifetime smoking performed by Wootton and colleagues detailed in Table C in S1 Data. The estimates of their association with MS are taken from the 2019 MS Chip IMSGC meta-analysis. CI, confidence interval; GWAS, genome-wide association study; IMSGC, International Multiple Sclerosis Genetics Consortium; MR, Mendelian randomization; MS, multiple sclerosis; OR, odds ratio; p.val, *p*-value; RAPS, robust adjusted profile score; SD, standard deviation; SNP, single nucleotide polymorphism.

MR methods employed (Fig 3 and Fig E in S1 Data). While the MR–Egger estimate deviated somewhat from that of the IVW and other sensitivity analysis, this was explained by its reduced power as evidenced by the wide CIs, which still overlapped the null (OR 1.34, 95% CI 0.49 to 3.65). There was evidence of heterogeneity among the individual SNP effect estimates for lifetime smoking with a large Cochran Q statistic (156.18, $p = 0.02$) and MR-PRESSO global test estimate of 158.4895, $p = 0.03$. However, this was not supported by the symmetrical funnel plot (Fig F in S1 Data) nor by any outliers detected in the MR-PRESSO test. Furthermore, the small MR–Egger intercept (−0.003, $p = 0.69$) suggests that the magnitude of potential bias from directional pleiotropy is low. Furthermore, there was no single SNP driving the association whose effect is being masked in the overall estimate as demonstrated by the leave-one-out and single-SNP sensitivity analyses (Fig G and H in S1 Data). MR excluding the lifetime smoking–associated variant located within the MHC region yielded consistent results overlapping the null (Table D in S1 Data).

A bidirectional analysis shows that there was no clear evidence that a genetic predisposition to MS is associated with either smoking initiation or lifetime smoking (Table E and F in S1 Data). A sensitivity MR of using MS-associated variants located within the MHC region yielded consistent results overlapping the null (Table G and H in S1 Data).

## Discussion

This study uses the MR method to estimate the causal effect of smoking on risk for MS. Using a two-sample MR design in 14,802 MS cases and 26,703 controls, we found little evidence that

both genetically predicted smoking initiation and lifetime smoking are associated with MS risk. These findings suggest that smoking is not a clear environmental risk factor for MS susceptibility and are in line with a recent independent study [27]. That study similarly found that smoking initiation and lifetime smoking were not associated with increased MS risk using a two-sample MR approach within the same MS susceptibility GWAS. The authors also investigated the association between smoking heaviness and MS, although the interpretation of this exposure in the absence of stratification by smoking status is unclear. Indeed, stratification in this context is important as the effect of the smoking heaviness, proxied by the genetic instrument, should be examined among current smokers only, not never smokers. An important difference with our study is the additional sensitivity analyses performed herein to assess potential bias from pleiotropy, which is especially important given that pleiotropy can only be tested indirectly. Indeed, we obtained MR estimates excluding smoking-related variants in the MHC region due to its high potential for pleiotropy, in line with previous MR studies in MS [28]. We also report results from MR-RAPS and MR-PRESSO analyses. Consistent estimates across these additional pleiotropy robust MR methods increase the validity of our findings. Moreover, our analysis included an increased number of variants as we used an $r^2$ threshold of 0.8 instead of 0.9 for identifying proxies and did not exclude palindromic variants (given that all genetic datasets are on the same genome build), resulting in slightly narrower CI. Although a small effect cannot be entirely excluded, the relatively narrow CIs, particularly for smoking initiation (0.92 to 1.16), make a clinically relevant effect less likely.

This contradicts previously reported observational studies that show an association with MS risk among smokers, compared to nonsmokers, of a meta-analysed effect estimate OR of 1.5 [4,11]. The studies included limitations such as self-report MS diagnosis [29], participation rate less than 80% [30–32], and loss to follow-up [33]. Additionally, observational studies may have heterogenous results due to how smoking status was defined [11]. The strength of association and causality between smoking and MS risk has been suggested due to a dose-dependent relationship in duration and intensity of smoking [4,34] as well as from the interaction between compounds present in cigarettes and specific genetic HLA variants, which include the presence of HLA-DRB1*15, the absence of HLA-A*0201 [35], and specific N-acetyltransferase 1 (NAT1) polymorphisms [36]. Smoking status may strongly influence the risk of developing MS associated with these genetic variants. This is thought to be through facilitation of epitope cross-reactivity and subsequent activation of T cells. However, other studies have failed to replicate this interaction [37,38]. In order to test this interaction in an MR casual inference context, a factorial MR design in MS patients with and without those alleles would be required. This was not possible in the present study due to the use of GWAS summary statistics. Observational estimates may have also been biased by residual or unmeasured confounding from factors influencing both smoking status and MS. For example, comorbidities and socioeconomic status may influence the likelihood of being a smoker and having MS [15,39].

In as much as we could, we ensured that there was no sample overlap between participants (case and controls from the MS GWAS) in the exposure and outcome GWAS by using consortiums that comprised completely separate cohorts. Indeed, the MS GWAS did not include the UK Biobank, ensuring no overlap between lifetime smoking and MS GWAS cohorts. There is a potential for overlap between the controls included in the smoking initiation GWAS, and those of the MS GWAS, however, will not lead to bias.

Reverse causality arises if preclinical aspects of a disease affect the risk factor; in this case, preclinical aspects of MS might influence the likelihood of a person smoking. This could partly explain the discrepancy between our MR results and observational studies especially as MS onset may occur long before the first clinical symptoms [39]. For instance, this prodromal phase is characterised in part by a higher risk of depression and anxiety up to 10 years prior to

MS diagnosis [40], and these, in turn, are associated with a higher rate of smoking. This study sought to reduce bias from confounding and reverse causation by using an MR design given genetic variants are much less associated with confounders than directly measured environmental exposures [41] (here smoking), and genetic variants are fixed over our lifetime-ensuring directionality of effect. This is a major strength of this study in establishing causality in the relationship between smoking and MS risk. Additionally, MR reverse direction was performed and showed that reverse causation is unlikely to be playing a role. A further strength of this study is the use of robust genetic instruments, which are strong predictors of smoking behaviour. Finally, we used multiple MR methods and sensitivity analyses to test for bias from directional horizontal pleiotropy. Our estimates were consistent across these multiple methods, strengthening our conclusions.

Smoking phenotypes are correlated with BMI, and BMI has a causal effect on MS susceptibility [42]. This is an interesting and complex relationship. Multivariable MR allows for the adjustment of 2 correlated exposures, here smoking and BMI, and investigating their effect on MS susceptibility and would be an interesting follow-up study. Vandebergh and colleagues used this framework to show that BMI but not smoking initiation had a causal effect on MS risk [27].

The current study cannot inform us about the effects of smoking on MS symptom severity, disability, or progression of disease. Indeed, smoking shows an association with disease progression, disease activity (new lesions on MRI and clinical relapse rates), and brain atrophy [15]. Observational studies have shown an association between smoking and progression from relapsing remitting MS to secondary progressive MS with a dose–response relationship [43–47] as well as a faster rate increasing Expanded Disability Status Scale (EDSS) [22]. However, more research in this area is needed for a definitive conclusion of an effect and specific mechanisms of action. As new methods are being developed to assess disease progression using MR [48], when a GWAS of MS progression becomes available, future studies should explore the association between smoking and the different measures of MS progression in an MR framework.

The instruments predicting smoking initiation and lifetime smoking were broadly distinct (only 9 SNPs overlapping). The measure of lifetime smoking exposure takes into account smoking status and, among ever smokers, duration, heaviness, and cessation. Although our lifetime smoking instrument captured smoking heaviness in part, however, we were unable to explore whether there was a dose–response relationship between the number of cigarettes smoked and the likelihood of developing MS in a two-sample MR context given that the MS GWAS is not stratified by smoking status. Most, but not all [31,49,50], evidence to date seems to suggest that there is a positive correlation between the amount smoked and the severity of illness [4,32,38,44,51–54]. It might be that rather than a causal relationship between smoking and MS risk, that smoking instead accelerates the disease process in those that would have already developed MS.

Limitations of this study are, firstly, that although we assessed pleiotropy using MR methods that account for pleiotropic effects, pleiotropy can only be addressed indirectly, and some SNPs may relate to MS risk through pathways other than smoking. We did not find evidence for bias for horizontal pleiotropy using the MR–Egger intercept test nor the funnel plots, which did not reveal evidence of directional, or unbalanced, pleiotropy. Secondly, this study was a two-sample MR using MS meta-analysis summary statistics, and therefore, this does not allow for gene–environment interaction or sex-stratified analysis.

In conclusion, we find no clear evidence for a causal effect of smoking on the risk of developing MS. Previous observational results may have been due to confounding factors, which we have avoided through our analysis. Future research should focus on the effect of smoking on the disease course of MS and its effect on progression.

## Methods

### Genetic instruments for smoking

**Smoking initiation.**   We used the most recent GWAS of smoking initiation from the GWAS & Sequencing Consortium of Alcohol and Nicotine use (GSCAN) consortium, which identified 378 conditionally independent genome-wide significant SNPs in a sample of 1,232,091 individuals of European ancestry. These genetic variants explain 2% of the variance in smoking initiation [25]. A previous study from our group has shown that a polygenic risk score for genetic variants for smoking initiation predict smoking behaviour (self-report smoking initiation and ever e-cigarette use) in the Avon Longitudinal Study of Parents and Children [55].

**Lifetime smoking.**   In order to incorporate measures of smoking heaviness without having to stratify on smoking status (which is not possible in the two-sample MR context without a stratified GWAS of MS), we used the GWAS of lifetime smoking conducted in 462,690 individuals of European ancestry from the UK Biobank [26]. Lifetime smoking is a combination of smoking initiation, duration, heaviness, and cessation described in detail elsewhere [26]. This GWAS identified 126 independent genome-wide significant SNPs that explain 0.36% of the variance [26]. This instrument, generated by our group, was validated using positive control outcomes of lung cancer, coronary heart disease, and hypomethylation at the aryl hydrocarbon receptor repressor site cg0557592 1 [26].

### Genetic variants associated with multiple sclerosis

Effect estimates and standard errors for smoking-associated SNPs on MS susceptibility were obtained from the summary statistics of the discovery cohorts of the latest International Multiple Sclerosis Genetics Consortium (IMSGC) meta-analysis, including 14,802 cases and 26,703 controls [56]. All details relating to demographic characteristics, MS case ascertainment, and eligibility criteria for the meta-analysis can be found in the original publication [56]. For SNPs not available in the IMSGC dataset, we identified proxy SNPs in high linkage disequilibrium ($r^2 > 0.8$) using an online tool LDlink (https://ldlink.nci.nih.gov/?tab=ldproxy), giving a total of 371 SNPs for smoking initiation instrument and 125 SNPs for lifetime smoking (Fig 1 and Table A and C in S1 Data).

**Mendelian randomization analyses.**   A two-sample MR was undertaken to obtain effect estimates of genetically predicted smoking on MS susceptibility, using both initiation and lifetime proxy measures. MR and sensitivity analyses were performed in R (version 3.5.1) using the TwoSampleMR R package (https://mrcieu.github.io/TwoSampleMR/) [57] with effect estimates compared across 5 different methods: IVW, MR–Egger [58], weighted median [59], weighted mode [60], RAPS [61] and PRESSO [62]. Given the different assumptions that each of these methods makes about the nature of pleiotropy, consistency in the point estimate across the methods strengthens causal evidence [63]. For instance, the MR–Egger method provides valid estimates even in the presence of pleiotropic effects as long as the size of these effects is independent of the effect of the genetic variants on the exposure (known as the INstrument Strength Independent of Direct Effect [InSIDE] assumption). The IWV method is the main analysis, and the other methods provide sensitivity analyses. Instrumental variable analysis of MR is based on a ratio of the regressions of the genetic instrument–outcome association (weighted smoking–associated SNPs with MS from IMSGC) on the genetic instrument–exposure association (smoking-associated SNPs with smoking initiation or lifetime smoking in the independent smoking GWASs). For smoking initiation, the ORs are expressed per unit increase in log odds of ever smoking regularly (smoking initiation), with a 1 standard

deviation (SD) increase in genetically predicted smoking initiation corresponding to a 10% increased risk of smoking [25]; for lifetime smoking, the ORs are expressed per 1 SD increase of the lifetime smoking index. An SD increase in the lifetime smoking score is equivalent to an individual smoking 20 cigarettes a day for 15 years and stopping 17 years ago or an individual smoking 60 cigarettes a day for 13 years and stopping 22 years ago [26].

Additional sensitivity analyses were performed in order to formally test for potential violations of MR assumptions. The mean F statistic was calculated as an indicator of instrument strength (a value of >10 indicates a strong instrument), and the Cochran Q statistic was assessed as a measure of heterogeneity for the IVW method to estimate whether the individual SNP effects of smoking on MS were inconsistent. The MR–Egger intercept was assessed to detect directional pleiotropy where the genetic instruments would be influencing MS through another pathway other than smoking. To identify potentially influential SNPs, which could be driven for example by horizontal pleiotropy, we used leave-one-out and single-SNP MR analyses. Additionally, due to the strong genetic signal for MS within the MHC region and high potential for pleiotropy, MR analysis excluding exposure variants located within the extended MHC region was performed (defined as base positions 24,000,000 to 35,000,000 on chromosome 6 [GRCh37]).

## Supporting information

**S1 Data. Supplementary figures and tables.**
(DOCX)

## Author Contributions

**Conceptualization:** Ruth E. Mitchell, Kirsty Bates, Robyn E. Wootton, George Davey Smith, Marcus R. Munafò.

**Formal analysis:** Ruth E. Mitchell, Kirsty Bates.

**Investigation:** Ruth E. Mitchell.

**Methodology:** Ruth E. Mitchell, Robyn E. Wootton, Adil Harroud.

**Project administration:** Ruth E. Mitchell, Robyn E. Wootton.

**Supervision:** George Davey Smith, Marcus R. Munafò.

**Writing – original draft:** Ruth E. Mitchell, Kirsty Bates, Robyn E. Wootton, Adil Harroud.

**Writing – review & editing:** Ruth E. Mitchell, Kirsty Bates, Robyn E. Wootton, Adil Harroud, J. Brent Richards, George Davey Smith, Marcus R. Munafò.

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
