## [Editor Report · Decision Letter 0]

25 Jun 2020

Dear Dr Mitchell, 

Thank you for submitting your manuscript entitled "The effect of smoking on multiple sclerosis: a mendelian randomization study" for consideration as a Research Article by PLOS Biology.

Your manuscript has now been evaluated by the PLOS Biology editorial staff as well as by an academic editor with relevant expertise and I am writing to let you know that we would like to send your submission out for external peer review.

Please re-submit your manuscript within two working days, i.e. by Jun 27 2020 11:59PM.

Kind regards,

Di Jiang, PhD

Senior Editor

PLOS Biology

---

## [Decision Letter · Decision Letter 1]

19 Aug 2020

Dear Dr Mitchell,

Thank you very much for submitting your manuscript "The effect of smoking on multiple sclerosis: a mendelian randomization study" for consideration as a Research Article at PLOS Biology and please accept my apologies for the time it has taken us to contact you with a decision on your study. Your manuscript has been evaluated by the PLOS Biology editors, an Academic Editor with relevant expertise, and by several independent reviewers whose comments you will find below (you will see that reviewer 1, Dipender Gill, has identified himself but provided an extremely uncritical report.

In light of the reviews (below), we are pleased to offer you the opportunity to address the issues raised by the reviewers in a revised version that we anticipate should not take you very long. We will then assess your revised manuscript and your response to the reviewers' comments and we may consult the reviewers again. It would be particularly important to pay attention to the issues raised by reviewer 4, who is the member of the panel with the expertise to gauge the MS epidemiological aspects of the work and the conclusiveness of many of the specific analyses done, as well as the most critical with the work.

We expect to receive your revised manuscript within 1 month, but please do let us know if you would need more time.

**IMPORTANT - SUBMITTING YOUR REVISION**

*Resubmission Checklist*

*Published Peer Review*

*PLOS Data Policy*

*Blot and Gel Data Policy*

With kind regards,

Nonia

Nonia Pariente, PhD,

Senior Editor,

npariente@plos.org,

PLOS Biology

REVIEWS:

Reviewer's Responses to Questions

PLOS authors have the option to publish the peer review history of their article (what does this mean?). If published, this will include your full peer review and any attached files.

Reviewer #1: Yes: Dipender Gill

Reviewer #2: No

Reviewer #3: No

Reviewer #4: No

Reviewer #1: A sounds study that is suitable for publication in its current form if the editors believe it to carry sufficient priority.

Reviewer #2: This is a revision of a manuscript describing the results of a study using mendelian randomization to study the effects of smoking on MS.

This is a well written paper and the results appear robust. However, I am not an expert in Mendelian randomization so I cannot speak to the veracity of the analytical methods employed.

I only have one suggested revision:

Line 205: why were you unable to stratify by smoking status? Please provide further explanation

Reviewer #3: 

Numerous observational studies and meta-analyses have suggested a causal relationship between smoking and MS risk, to the extent that it is considered a true risk factor amongst clinicians and MS researchers. 

In this paper Mendelian randomization (MR) is used to assess the impact of smoking on the risk of MS. The critical assumptions as stated are: 1) the genetic variants robustly predict the exposure, 2) the genetic variants must not be associated with any confounders and 3) the genetic variants must only affect the outcome through the exposure.

While the results are negative, i.e. the authors find that there is no association between smoking and MS risk using MR, they are nonetheless important since this concept of smoking increasing risk of MS is firmly intrenched. 

The paper is overall well-written, and the figures are displayed properly. The statistical methods are sound, and the conclusions follow from the results.

It strikes me that the manuscript would benefit from slightly greater discussion of the Vandebergh study, ref 27, not simply as support but also comparing and contrasting methodology and sample populations.

The major criticism I have is that a positive control for the analysis would strengthen the conclusions. Please consider.

May I ask the authors to please expand on the validity or lack of validity of an odds ratio of 1.34 (figure 4). With the confidence interval shown the p value of .56 doesn't immediately make sense.

Also based on: Burgess S, Thompson SG. Interpreting findings from Mendelian randomization using the MR-Egger method Eur J Epidemiol. 2017;32(5):377-389. doi:10.1007/s10654-017-0255-x, may I ask the authors to comment on the idea:

"While the MR-Egger method is a worthwhile sensitivity analysis for detecting violations of the instrumental variable assumptions, there are several reasons why causal estimates from the MR-Egger method may be biased and have inflated Type 1 error rates in practice, including violations of the InSIDE assumption and the influence of outlying variants. The issues raised in this paper have potentially serious consequences for causal inferences from the MR-Egger approach." 

For this section of the discussion: "Reverse causation could also partly explain the discrepancy between our MR results and 176 observational studies especially as MS onset may occur long before the first clinical 177 symptoms (39). For instance, this prodromal phase is characterized in part by a higher risk of 178 depression and anxiety up to 10 years prior to MS diagnosis (40), and these in turn are 179 associated with a higher rate of smoking. This study sought to reduce bias from confounding 180 and reverse causation by using a MR design given genetic variants are much less associated 181 with confounders than directly measured environmental exposures (41) (here smoking) and 182 genetic variants are fixed over our lifetime ensuring directionality of effect. This is a major 183 strength of this study in establishing causality in the relationship between smoking and MS 184 risk.", can you please be more explicit about how/why observational studies and meta-analyses in particular are prone to reverse causation.

Minor:

In addition, cigarette chemicals contribute mechanistically to MS…… 

In addition, cigarette chemicals may contribute mechanistically to MS…… 

Recommend not abbreviating Instrumental variable assumptions to IV.

Poorly worded: "Furthermore, MR-PRESSO removes individual SNPs that contribute to heterogeneity disproportionately more than expected in order to reduce heterogeneity."

Reword: "These genetic variants facilitated epitope cross-reactivity and activation of T cells and smoking may strongly influence the risk of MS observed with these HLA genotypes." These genetic variants are thought to influence MS risk by facilitating epitope cross reactivity but this is an assumption. 

Reviewer #4: Observational studies have suggested that smoking is a risk factor for multiple sclerosis (MS). This paper used a two-sample Mendelian randomization approach to evaluate the causal relationship between smoking and MS risk. They used the most recent IMSGC summary statistics for MS risk and summary statistics from GWAS for lifetime smoking and smoking initiation. They do not find evidence of an effect of smoking on the risk of MS and conclude that it is unlikely that smoking is a contributor to MS. While I think this study was generally well done, I do have some questions about the methodology and rigor of the analysis as well as some concerns related to the conclusions of the study. 

Did the other smoking phenotypes in the most recent smoking GWAS demonstrate similar effects (e.g. cigarettes per day) or conduct analyses assessing the association between age at smoking initiation and MS risk or age at MS onset or other smoking phenotypes like cotinine? Some MS risk factors may be particularly detrimental at certain age groups. Also, I realize that if only one of these additional phenotypes showed an effect, it may be a chance finding, but I think it would help the support the study's conclusions if a more comprehensive set of smoking phenotypes was considered. The study would also benefit if some justification of the specific phenotypes selected was provided. 

Are there overlapping participants in both the MS and smoking GWAS's (at least some of the controls from the MS study in the most recent GWAS? This should be at least discussed. 

Did the authors consider any interaction analyses with BMI, given that some smoking phenotypes are associated with BMI, though I realize this relationship is complex (as it could be for smoking/MS, but this is not considered or evaluated). 

While I agree that this analysis does not support smoking as, the MR confidence intervals are still quite large it's possible that the studies of MS are not large enough to detect such an effect.

Is the confidence interval for smoking cessation in the abstract correct? 1.03 (0.92, 1.61) - either the lower or upper bound seem incorrect. 

This is minor, but I think it would be helpful to soften some of the claims in the introduction. For example, it's not definitive that there is a genetic interaction between HLA risk alleles, smoking and MS. As the authors mention in the Discussion, this was not replicated in other populations, so perhaps this should be revised here.

---

## [Editor Report · Decision Letter 2]

8 Oct 2020

Dear Dr Mitchell,

Thank you for submitting your revised manuscript entitled "The effect of smoking on multiple sclerosis: a mendelian randomization study" for publication in PLOS Biology. I have had time to assess your revision and discussed it with the Academic Editor. 

We're delighted to let you know that we're now editorially satisfied with your manuscript and would like to proceed to publication of your study as a Short Report. However before we can formally accept your paper and consider it "in press", we also need to ensure that your article conforms to our guidelines. In going through your manuscript, we have noted the following formatting and reporting issues that will need to be addressed:

1) The figures need to be cited in order in the text, so some relabelling is needed. We would recommend to merge figures 1 and 2 into a 3-panel figure.

2) The Supplementary Figures need to have a legend (not only a title) that should be a detailed description of what is represented and include all relevant statistical information to understand the analysis performed.

3) You may be aware of the PLOS Data Policy, which requires that all data underlying the figures be made available without restriction: http://journals.plos.org/plosbiology/s/data-availability. For more information, please also see this editorial: http://dx.doi.org/10.1371/journal.pbio.1001797

Note that we do not require all raw data. Rather, we ask that all individual quantitative observations that underlie the data summarized in the figures and results of your paper be made available and that the figure legends in your manuscript include information on where the underlying data can be found. Underlying data can be provided in one of the following forms:

a) Supplementary files (e.g., excel). Please ensure that all data files are uploaded as 'Supporting Information' and are invariably referred to (in the manuscript, figure legends, and the Description field when uploading your files) using the following format verbatim: S1 Data, S2 Data, etc. Multiple panels of a single or even several figures can be included as multiple sheets in one excel file that is saved using exactly the following convention: S1_Data.xlsx (using an underscore).

b) Deposition in a publicly available repository. Please also provide the accession code or a reviewer link so that we may view your data before publication. 

c) In existing Supplementary Tables. If this is the case, please specify this in the relevant figure legend (e.g. source data for this figure can be found in Supp Table X).

Regardless of the method selected, please ensure that you provide the individual numerical values that underlie the data displayed in all of the main and supplementary figures, which is essential for readers to assess your analysis and to reproduce it.

Please ensure your source data file/s has a legend and the information is presented in an accessible way, such that readers can readily identify the data and understand how the figure was generated.

Please ensure that your Data Statement in the submission system accurately describes where your data can be found; i.e. includes a mention to the supplemental data files or to a repository (with accession number).

---

In addition, a member of our team will be in touch shortly with a different set of requests to ensure your manuscript adheres to our policies. As we can't proceed until these requirements are met, your swift response will help prevent delays to publication. 

- a cover letter that should detail your responses to any editorial requests, if applicable

- a track-changes file indicating any changes that you have made to the manuscript, if applicable

We expect to receive your revised manuscript within two weeks. Please let us know if this final revision is likely to take longer.

*Copyediting*

*Published Peer Review History*

*Early Version*

With best wishes,

Nonia

Nonia Pariente, PhD,

Editor-in-Chief,

npariente@plos.org,

PLOS Biology

---

## [Editor Report · Decision Letter 3]

29 Oct 2020

Dear Dr Mitchell,

On behalf of my colleagues and the Academic Editor, Richard Daneman, I am pleased to inform you that we will be delighted to publish your Short Reports in PLOS Biology. 

PRODUCTION PROCESS

Before publication you will see the copyedited word document (within 5 business days) and a PDF proof shortly after that. The copyeditor will be in touch shortly before sending you the copyedited Word document. We will make some revisions at copyediting stage to conform to our general style, and for clarification. When you receive this version you should check and revise it very carefully, including figures, tables, references, and supporting information, because corrections at the next stage (proofs) will be strictly limited to (1) errors in author names or affiliations, (2) errors of scientific fact that would cause misunderstandings to readers, and (3) printer's (introduced) errors. Please return the copyedited file within 2 business days in order to ensure timely delivery of the PDF proof. 

If you are likely to be away when either this document or the proof is sent, please ensure we have contact information of a second person, as we will need you to respond quickly at each point. Given the disruptions resulting from the ongoing COVID-19 pandemic, there may be delays in the production process. We apologise in advance for any inconvenience caused and will do our best to minimize impact as far as possible.

EARLY VERSION

PRESS 

Kind regards,

Alice Musson

Publishing Editor, 

PLOS Biology

on behalf of

Nonia Pariente, PhD,

Editor-in-Chief

PLOS Biology